# TraDiffusion++: Hierarchical Guidance for Fine-Grained Trajectory-Based Image Generation

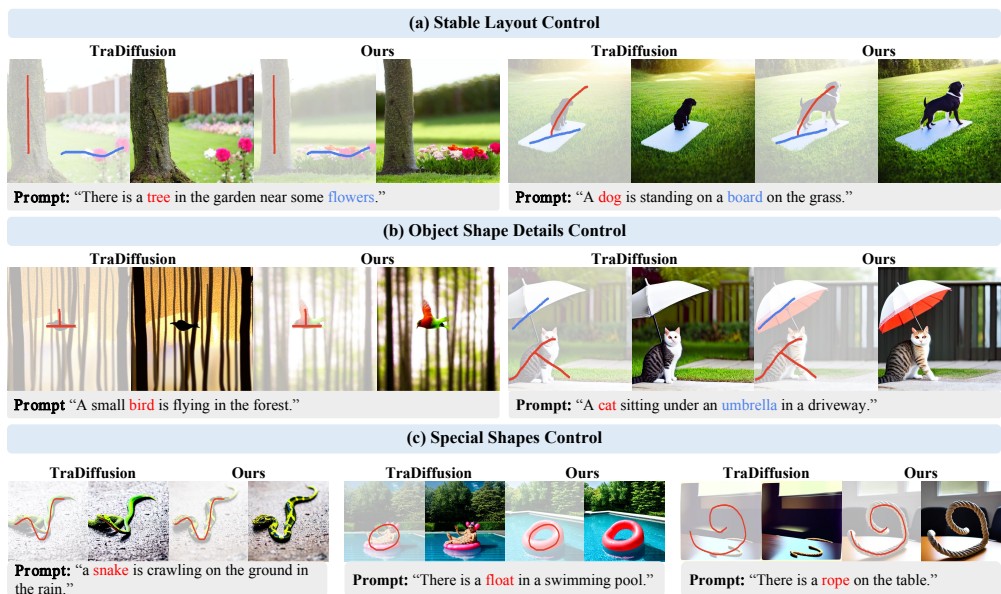

Figure 1: **Controllable text-to-image synthesis with trajectories.** Compared to TraDiffusion, our method offers more stable layout control. Furthermore, it achieves fine-grained control over objects, such as generating object shape details and special shape objects.

## ABSTRACT

Currently, many training-free methods based on diffusion models allow controllable generation. These methods, such as TraDiffusion, introduce control through additional trajectory input. While they are more user-friendly than traditional methods, they offer only coarse control over the Stable Diffusion (SD) model. We observe that SD focuses more on layout control at lower resolutions of cross-attention and shape control at higher ones. Based on this, we propose TraDiffusion++, which introduces a Hierarchical Guidance Mechanism (HGM) for finer-grained control in generation. HGM includes three key components: Control Loss (CL), Suppress Loss (SL), and Fix Loss (FL). CL aligns the layout with the trajectory across layers. SL suppresses objects outside the trajectory at lower resolutions. FL refines regions not fully controlled by the trajectory using attention feedback at middle and high resolutions. The combination of CL and SL ensures effective layout control. The interaction between CL and FL improves shape generation. We build a dataset with simple and complex trajectories. Experiments show that TraDiffusion++ achieves stable layout control and fine-grained object generation. This also reveals new insights into SD's control mechanisms.

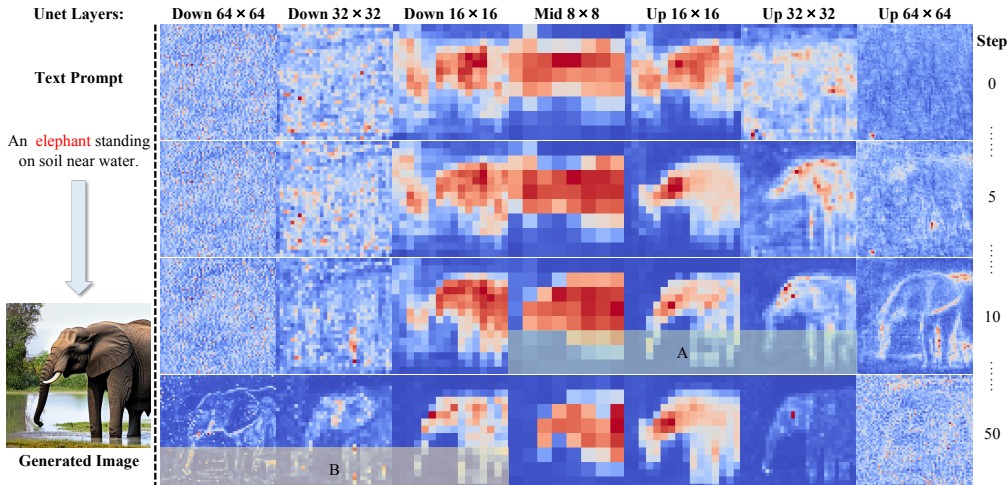

Figure 2: **Analysis of cross-attention maps at different resolutions in StableDiffusion.** The experiment, based on SD-v1.5, generates images from given prompts. SD's U-Net structure includes (Down), (Mid), and (Up) layers, each with cross-attention at varying resolutions. We visualize the cross-attention maps for the token "elephant" at time steps 0, 5, 10, and 50 across different resolutions.

# 1 INTRODUCTION

In recent years, text-to-image models trained on large-scale datasets Ramesh et al. (2021; 2022); Rombach et al. (2022) have made significant advances in image generation. Text prompts provide a flexible way to guide generation, but there is a gap between text and images. This gap often prevents the generated images from fully aligning with the text prompts. It also limits the ability to specify details like object position or shape.

To overcome these limitations, some models have introduced generalized training methods based on existing text-to-image models Zhang et al. (2023); Mou et al. (2024); Li et al. (2023). These approaches use additional visual conditions to control image generation, achieving notable improvements. However, they come with high training costs. More recently, training-free control methods have emerged. These methods guide pre-trained diffusion models using energy functions Phung et al. (2024); Chen et al. (2024b); Xie et al. (2023); Kim et al. (2023b). Examples include using object masks Couairon et al. (2023) or bounding boxes Xie et al. (2023); Chen et al. (2024b); Phung et al. (2024). Despite this, traditional visual control methods are often not user-friendly. Masks are too detailed and hard to create, while boxes are too coarse and cannot precisely define object shapes. TraDiffusion Wu et al. (2024) addresses this by introducing trajectory-based control, offering a simpler way to guide image generation. However, as shown in Figure 1 (a), TraDiffusion struggles with stable control using simple trajectories. It is also limited to layout control and cannot handle complex trajectories. As seen in Figure 1 (b) and (c), it fails to generate detailed shapes or special-shaped objects.

To overcome the above limitations, we first perform an in-depth analysis of the architecture of Stable Diffusion (SD). By visualizing the cross-attention maps in SD's U-Net at different resolutions, we observe distinct behaviors: lower resolutions focus on layout generation, while higher resolutions capture finer object shapes. TraDiffusion controls the cross-attention maps only at the 8×8 and 16×16 resolution layers, but it does not fully utilize the unique properties of the different resolution cross-attention maps, leading to rough control over the layout and neglecting fine-grained object generation.

Building on these insights, we propose TraDiffusion++, a trajectory-based method for precise, controllable image generation without the need for retraining. Like TraDiffusion, our approach guides latent representations using energy functions during the denoising process. However, TraDiffusion++ introduces a Hierarchical Guidance Mechanism (HGM) that targets different resolutions of the cross-attention maps. This mechanism includes two key modules: Layout Guidance for low-

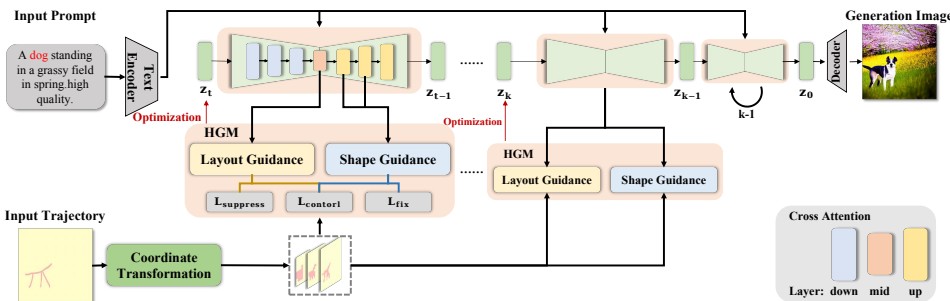

Figure 3: **Overview of Hierarchical Guidance Mechanism (HGM).** Given an input prompt and object trajectory, the object trajectory is transformed through coordinate transformation to the same resolution as the controlled cross-attention maps, serving as the control area. During the denoising process, gradient optimization of latent representations is performed using Layout Guidance and Shape Guidance to achieve fine-grained control over the object generation. Layout guidance consists of Control Loss and Suppress Loss, while shape guidance comprises Control Loss and Fix Loss.

resolution control and Shape Guidance for higher-resolution shape refinement. To implement this, we design three loss functions: Control Loss (CL), Suppress Loss (SL), and Fix Loss (FL). CL ensures that the layout aligns with the trajectory across different resolution layers, while SL operates at lower resolutions to suppress the generation of objects outside the trajectory. FL refines the control in the middle and high resolutions, using attention feedback to recover areas not fully controlled by the trajectory. Together, CL and SL guarantee stable layout control, while the combination of CL and FL enables precise shape generation. We further refine TraDiffusion by adapting trajectory coordinates to different resolution layers, preventing boundary blurring and ensuring more stable and accurate layout generation. This multi-resolution strategy enhances fine-grained object control, offering better fidelity and detail in the generated images.

Through extensive qualitative and quantitative evaluations, TraDiffusion++ demonstrates superior control over layout and shape generation compared to existing methods. Our analysis and experimental results validate the effectiveness of our approach, revealing new insights into SD's control mechanisms and significantly improving image quality.

Our contributions can be summarized as follows:

- Building on our analysis of SD's mechanism, we propose a new training-free approach that adapts text-to-image models for trajectory-based control.

- We design HGM, which integrates Control Loss, Suppress Loss, and Fix Loss to effectively manage layout at lower resolutions and achieve fine-grained shape control at higher resolutions.

- We construct a novel dataset containing objects with simple and complex trajectories and introduce the IoT metric to measure whether the generated objects are accurately aligned with their corresponding trajectories.

## 2 RELATE WORK

**Text-to-Image Diffusion Models.** With the emergence of large-scale diffusion models, these models Rombach et al. (2022); Ramesh et al. (2022); Saharia et al. (2022); Nichol et al. (2021) have achieved remarkable results in text-to-image tasks by progressively adding noise to images and learning to denoise them. LDM Rombach et al. (2022) improves computational efficiency by compressing images into a latent space, allowing the model to capture essential information. DALL·E 2 Ramesh et al. (2022) integrates CLIP's image space, using contrastive learning to make generated images better match text descriptions. At the same time, recent research indicates that using classifier-free guidance Ho & Salimans (2022) can effectively improve the alignment between the generated images and the text prompts.

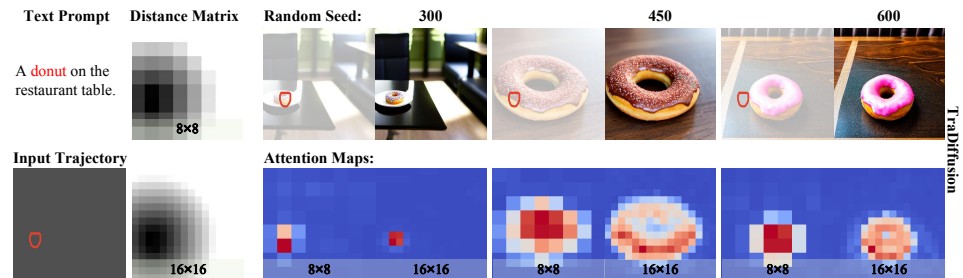

Figure 4: **Visualization of TraDiffusion's inability to stably control layouts.**

**Controllable Text-to-Image Generation.** Current controllable text-to-image generation methods address issues like context understanding, entity loss, and attribute leakage by introducing additional conditions such as masks, bounding boxes, or depth maps, leading to images that better meet user expectations. Recent innovations Zhang et al. (2023); Mou et al. (2024); Li et al. (2023); Qin et al. (2023); Kim et al. (2023a); Chen et al. (2024a); Huang et al. (2023); Avrahami et al. (2023); Yang et al. (2023); Luo et al. (2024); Koley et al. (2024); Ju et al. (2023); Wang et al. (2024); Hu et al. (2024); Voynov et al. (2023) use pre-trained text-to-image models and additional trainable modules to achieve controllable generation. For example, ControlNet Zhang et al. (2023) achieves significant results by integrating knowledge into Stable Diffusion through zero convolution operations. However, these methods often require costly computational resources and extensive data. Newer approaches Chen et al. (2024b); Kim et al. (2023b); Xie et al. (2023); Wu et al. (2024); Phung et al. (2024); Mo et al. (2024); Couairon et al. (2023); Zhao et al. (2023) address this by designing energy functions to guide the diffusion process and optimizing cross-attention maps or feature maps during denoising for efficiently controllable image generation.

## 3 METHOD

### 3.1 PRELIMINARIES

**Stable Diffusion model.** Our method is based on Stable Diffusion (SD) Rombach et al. (2022) model, which is primarily composed of a text encoder, image encoder, image decoder, and denoising network U-Net Ronneberger et al. (2015). The U-Net is divided into three parts: downsampling, middle, and upsampling layers. Unlike traditional diffusion models, SD enhances computational efficiency by compressing images into latent representations. Simultaneously, it facilitates text-to-image generation by converting text prompts into fixed-length embeddings using a text encoder. These embeddings are then fused with latent representations at various resolutions and levels through a cross-attention mechanism, which can be formulated as follows:

$$A = \mathrm{softmax}(\frac{Q \cdot K}{\sqrt{d_k}}), \tag{1}$$

where Q is a linear transformation of the latent representations, and K is from text embeddings. The resulting A reflects the degree of association between the visual information at a specific position and the corresponding text information.

**TraDiffusion model.** TraDiffusion is a training-free, trajectory-based, controllable generation method built on the Stable Diffusion model. Given an input prompt $y$ and $n$ control objects $\{(l_1, T_1), (l_2, T_2), \ldots, (l_n, T_n)\}$, where $l_i$ represents the object label and $T_i$ represents the corresponding object trajectory, it aims to control object generation using simple trajectories. Specifically, it converts the object trajectory into a distance matrix and then downsampling it to the same resolution as the controlled cross-attention maps, it serves as the control area. It optimizes latent representations through gradient backpropagation using control and movement losses, guiding the cross-attention map values to focus on the control area. This process ensures alignment between the object and its trajectory, which can be formulated as follows:

$$z_t \leftarrow z_t - \sigma_t^2 k \nabla_{z_t} \sum_{\eta \in \delta} \sum_{i \in \mathbb{N}} E\left(A^{(\eta)}, T_i, l_i\right), \tag{2}$$

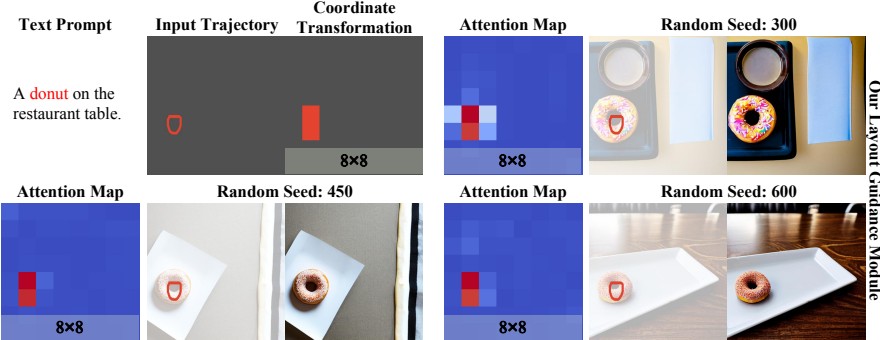

Figure 5: **Visualization of Our Layout Guidance Module's ability to stably control layouts.**

where $z_t$ represents the latent representations at time step $t$, $A^{(\eta)}$ represents the cross-attention maps of the $\eta$-th layer in the U-Net, $k > 0$ is a scale factor that adjusts the guidance strength, $\mathbb{N} = \{1, \cdots, n\}$, $\delta$ is the set of controlled layers, and $\sigma_t = \sqrt{\frac{1-\alpha_t}{\alpha_t}}$, with $\alpha_t$ is a predefined coefficient that controls noise attenuation or scaling Rombach et al. (2022). We find that TraDiffusion cannot stably control the object layout because it simulates the object mask using a distance matrix centered around the object trajectory, with values increasing with distance, as shown in Figure 4. However, this large control area does not effectively stabilize the generation of the object layout. For example, when the random seed changes, as illustrated in Figure 4, it struggles to maintain a stable layout of the object "donut". Additionally, it lacks the ability for fine-grained control over the objects.

### 3.2 LAYERS ATTENTION ANALYSIS

Previous works Chen et al. (2024b); Wu et al. (2024) only utilize the strong layout correspondence between Stable Diffusion (SD) cross-attention maps and the final generated images to achieve layout-controllable generation, but they lack an in-depth analysis of how SD gradually generates object details during the denoising process. To realize fine-grained control of objects based on trajectories, based on the SD-v1.5 model, we use 50 denoising steps to generate object images and visualize cross-attention maps at different time steps and layers (as shown in Figure 2).

We find that, in the early stages of SD denoising, the cross-attention maps of Unet's middle and upsampling layers show a clear correspondence with the final generated image (as shown in Figure 2 A). In contrast, the downsampling layers' correspondence tends to appear later in the denoising process (as shown in Figure 2 B). Additionally, we find that in the low-resolution attention maps, this correspondence is reflected in the position of the object content, while as the resolution of cross-attention maps increases, the object shape details become more pronounced. For example, from Figure 2 A), at an 8x8 resolution cross-attention map, only the position of the elephant in the final generated image can be identified; however, with higher-resolution ones, the outline and curvature of the elephant's trunk gradually become more distinct. Similarly, while the details of the elephant's feet are not visible at low-resolution cross-attention maps, the boundaries become clear at high-resolution cross-attention maps. Crucially, these details are established early in the denoising process, where high response areas of cross-attention maps focus on the object shape's core regions.

Based on this, we summarize that, during the SD image generation process, it controls the generation of object layout at low resolutions, while the details of the object shape are primarily regulated at high resolution.

### 3.3 APPROACH OVERVIEW

Based on the above analysis, we propose TraDiffusion++ (as shown in Figure 3), a fine-grained controllable trajectory-based image generation method by redesigning TraDiffusion Wu et al. (2024). In Section 3.3.1, we detail the **Hierarchical Guidance Mechanism (HGM)** based on Section 3.2. This mechanism includes a **Layout Guidance Module (LGM)** and a **Shape Guidance Module (SGM)**, for which we design three loss functions: **Control Loss (CL)**, **Suppress Loss (SL)**, and **Fix Loss (FL)**. The LGM controls the generation of object layouts at low-resolution cross-attention

Figure 6: **Analysis of adding Attention Guidance to the 16x16 resolution upsampling layer.** We conduct a detailed analysis of the effects of different losses in controlling the 16x16 resolution cross-attention maps. Finally, (d) shows that combining Control Loss and Fix Loss can effectively manage the fine-grained generation of the object.

maps using CL and SL, while the SGM regulates the generation of object shapes at higher ones through CL and FL.

### 3.3.1 DESIGN OF HIERARCHICAL GUIDANCE

**Design of Layout Guidance Module (LGM).** As discussed in Section 3.1, the distance matrix-based approach in Tradiffusion leads to unstable layout control, so we apply coordinate transformation to convert the object trajectory to the same resolution as the controlled cross-attention maps (as shown in Figure 5) and redesign a LGM to control object layout generation. Similar to previous work Chen et al. (2024b), to ensure that the object is accurately generated within the specified area, we design a Control Loss function, which can be formulated as:

$$L_c \left( A^{(\eta)}, T_i, l_i \right) = \sum (1 - \frac{\sum \tilde{T}_i A^{(\eta)}_{pos(l_i)}}{\sum A^{(\eta)}_{pos(l_i)}}), \tag{3}$$

where $\tilde{T}_i$ denotes the control region transformed by $T_i$ coordinates to match the resolution of $A^{(\eta)}_{pos(l_i)}$, and $pos(l_i)$ is the index for calculating the control token $l_i$ in the cross-attention maps. To avoid the cross-attention maps of the object token from focusing excessively on unnecessary areas, which could lead to disorganized object generation or multiple unwanted repetitions, we design a Suppress Loss , which can be formulated as:

$$L_s \left( A^{(\eta)}, T_i, l_i \right) = (\frac{\sum (1 - \tilde{T}_i) A^{(\eta)}_{pos(l_i)}}{\sum A^{(\eta)}_{pos(l_i)}}). \tag{4}$$

The final LGM energy function can be formulated as follows:

$$E_{layout} \left( A^{(\eta)}, T_i, l_i \right) = L_c + L_s. \tag{5}$$

As shown in Figure 5, our LGM can stably control the object's layout generation.

**Design of Shape Guidance Module (SGM).** Based on our analysis in Section 3.2, the Layout Guidance Module (LGM) on the 8x8 resolution cross-attention maps cannot control the generation of object shapes because, at low resolutions, the cross-attention maps only correspond to the layout of the final generated object and cannot represent the object shape, as shown in Figure 6 (a). After adding the same loss used for the LGM to control the 16x16 resolution cross-attention maps, the object shape is controlled, as illustrated in Figure 6 (b). However, the generated object appears unnatural and overly conforms to the trajectory. This is due to using the Suppress Loss (SL) in controlling the 16x16 resolution cross-attention maps. According to our analysis in Section 3.2, the 16x16 resolution cross-attention maps have a strong correspondence with the shape of the final generated object, which significantly differs from the shape of our trajectory control region. The SL restricts the object shape from overly fitting our trajectory area, leading to distortion. Furthermore, our shape control objective focuses on guiding the core area of the object shape through trajectory

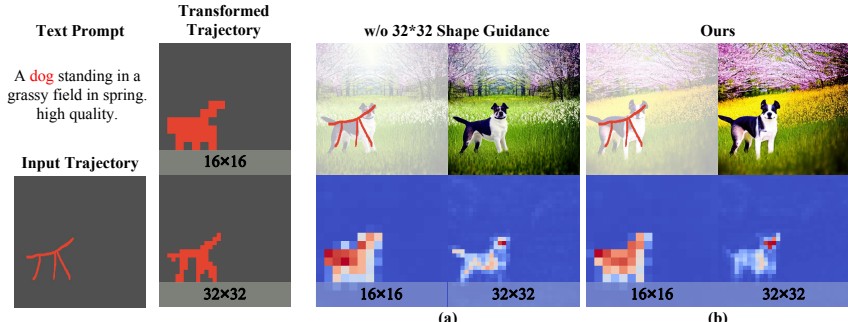

Figure 7: **Visualization of the effect of adding the Shape Guidance Module (SGM) over the 32x32 resolution upsampling layers.** By comparing (a) and (b), it is evident that adding the 32x32 SGM effectively improves the fine-grained generation capability of objects.

control rather than capturing the overall shape that includes all object details. However, simply relying on the Control Loss (CL) cannot accurately ensure the consistency between the object shape and the trajectory, as shown in Figure 6 (c). This is because the CL struggles to fully cover the entire control area, often resulting in losing part of the control region.

To address this, we design a Fix Loss that dynamically identifies core regions in the guided cross-attention maps and compares them with the control regions during the denoising process, filling in any missing parts as needed, which can be formulated as:

$$L_f\left(A^{(\eta)}, T_i, l_i\right) = (1 - \frac{\sum(\tilde{T}_{pos(l_i)}(\neg V_{pos(l_i)}))A^{(\eta)}_{pos(l_i)}}{\sum A^{(\eta)}_{pos(l_i)}}), \tag{6}$$

where $V_{pos(l_i)}$ is a binary mask dynamically generated before each guidance step by extracting high response regions from $A^{(\eta)}_{pos(l_i)}$. Specifically, the value of $V_{pos(l_i)}$ is set to 1 when the value at the corresponding position $A^{(\eta)}_{pos(l_i)}$ exceeds the threshold $u$; otherwise, it is set to 0. The final SGM energy function can be formulated as follows:

$$E_{shape}\left(A^{(\eta)}, T_i, l_i\right) = L_c + L_f. \tag{7}$$

As shown in Figure 6 (d), our SGM can effectively control the generation of object shapes. Additionally, based on our analysis in Section 3.2, the 32x32 resolution cross-attention maps have better shape representation capability, so we increase the control of the 32x32 resolution cross-attention maps to enhance shape control ability.

**The Energy Function of Hierarchical Guidance Mechanism (HGM).** Combining Layout Guidance Module and Shape Guidance Module, we design the energy function of the HGM, which can be formulated as follows:

$$E(A^{(\eta)}, T_i, l_i) = \lambda_1 E^{8\times 8}_{layout} + \lambda_2 E^{16\times 16}_{shape} + \lambda_3 E^{32\times 32}_{shape}, \tag{8}$$

where $\lambda_1$, $\lambda_2$, $\lambda_3$, are scale factors that adjust the guidance strength. Finally, we update the latent representations through backpropagation, which can be formulated as follows:

$$\boldsymbol{z}_t \leftarrow \boldsymbol{z}_t - \sigma_t^2 \nabla_{\boldsymbol{z}_t} \sum_{\eta \in \delta} \sum_{i \in \mathbb{N}} E\left(A^{(\eta)}, T_i, l_i\right), \tag{9}$$

where $\delta$ is the set of controlled layers, including the $8 \times 8$ resolution middle layers, the $16 \times 16$ resolution upsampling layers, and the $32 \times 32$ resolution upsampling layers.

## 4 EXPERIMENTS

### 4.1 EXPERIMENT SETUP

**Implementation Details.** Following previous works Wu et al. (2024), we conduct experiments based on the pre-trained text-to-image model SD-v1.5 Rombach et al. (2022). We compute the

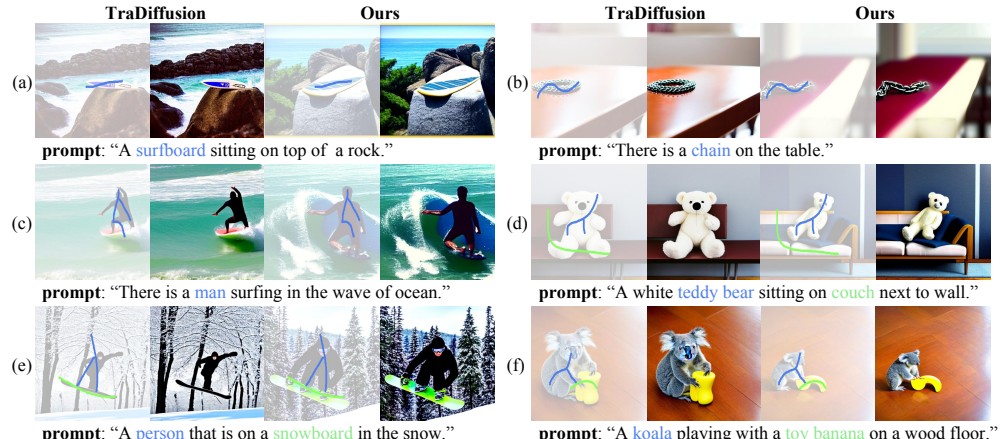

Figure 8: **Qualitative results of comparison with TraDiffusion.** From (a) to (f), our comparison gradually extends from simple trajectory control to complex trajectory control. Compared to TraDiffusion, we achieve more stable layout generation that aligns with simple trajectories while also maintaining fine control over objects in complex trajectories.

Table 1: **Qualitative results of comparison with prior work.** Simple and Complex correspond to our method to achieve the best performance in both DTL and IOT, particularly in the complex trajectories task. This demonstrates that our approach effectively aligns control trajectories and provides fine-grained control over the object.

| DataSets → | Simple | | | Complex | | |
|---|---|---|---|---|---|---|
| Method | IOT(↑) | DTL(↑) | Fid(↓) | IOT(↑) | DTL(↑) | Fid(↓) |
| Stable Diffusion | 0.26 | 0.0042 | 68 | 0.25 | 0.0039 | 59 |
| TraDiffusion | 0.53 | 0.0149 | 67 | 0.56 | 0.0186 | 55 |
| Ours | 0.62 | 0.0184 | 71 | 0.68 | 0.0230 | 58 |

energy function using cross-attention maps at the middle and upsampling layers across various resolutions. Images are generated over 50 denoising steps, with the energy function recalculated 5 times per step for the first 10 steps to update the latent representations. In our energy function, the hyperparameters are set as $\lambda_1 = 5$, $\lambda_2 = 20$, and $\lambda_3 = 15$, with a fixed random seed of 450.

**Evaluation Benchmark.** Following the setup of TraDiffusion Wu et al. (2024), we evaluate our method on the COCO2014 dataset Lin et al. (2014). We randomly select 1,000 images from the training set to create both a simple and a complex trajectory dataset, with each image containing 1-3 objects. In the simple trajectory dataset, each object's trajectory is represented by a single curve, while in the complex trajectory dataset, each object's trajectory includes 1-2 branches. The detailed construction process is further described in the Appendix. Since our method emphasizes solving the problem of fine-grained object control, we construct a unified dataset with 500 simple trajectory examples and 1,000 complex trajectory examples, totaling 1,500 examples, named "TRAT", for ablation studies.

**Evaluation Metrics.** FID Heusel et al. (2017) measures the quality of image generation by comparing the similarity of the real distributions of generated images and real images, while trajectory alignment is evaluated using DTL (Distance to Line) Wu et al. (2024). A higher DTL indicates better alignment, but it does not account for accurate object generation. If the object is poorly generated, DTL may still appear deceptively high. To address this, we introduce IOT (Intersection Over Trajectory), inspired by Accuracy Redmon et al. (2016) and IOU Everingham et al. (2010), which checks the correctness of object generation by comparing the trajectory with the object mask and calculating their overlap ratio. For this evaluation, we use YOLOv8x-Seg Redmon et al. (2016); Jocher et al. (2023) to obtain the object mask.

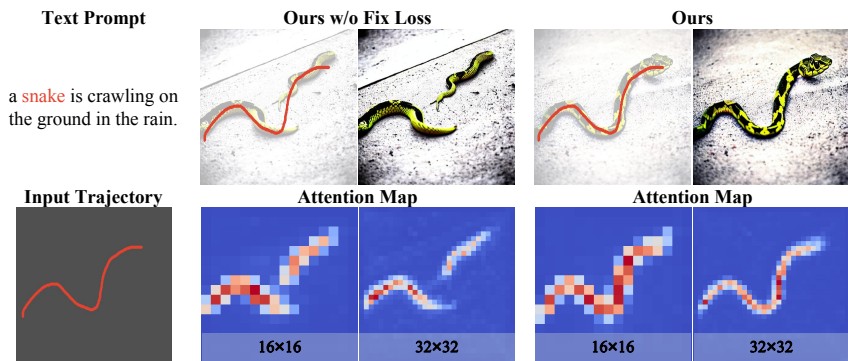

Figure 9: **Qualitative ablation study of Fix Loss (FL).** It indicates that under the influence of FL, we can effectively control the areas of cross-attention map loss during the guidance process, thereby resolving the issue of incoherent object generation.

Table 2: **Ablation of the Component of the Hierarchical Guidance.**

| Component | 8x8 Layout Guidance | 16x16 Shape Guidance | 32x32 Shape Guidance | IOT(↑) | DTL(↑) | FID(↓) |
|---|---|---|---|---|---|---|
| 1 | ✓ | ✗ | ✗ | 0.50 | 0.0098 | 58 |
| 2 | ✓ | ✓ | ✗ | 0.65 | 0.0212 | 60 |
| 3 | ✓ | ✓ | ✓ | 0.67 | 0.0214 | 61 |

## 4.2 COMPARISON WITH PRIOR WORK

We compare our method with the previous TraDIffusion Wu et al. (2024) approach.

**Qualitative Results.** We show examples of comparing our method with Tradiffusion on the simple and complex trajectories. As shown in Figure 8, our method demonstrates more stable performance in matching simple trajectories, such as the surfboard in (a) and the chain in (b). In the generation of complex trajectories, our method allows for more refined control over the objects, such as successfully generating human posture in (c). In multi-object generation scenarios with complex trajectories, we are also able to control the finer shape details of the objects, while TraDiffusion only generates the objects roughly around the given trajectory. This is evident in the shape of the teddy bear in (d), the human footsteps in (e), the koala's action, and the curvature of the banana in (f). In addition, as shown in Figure 10, our method demonstrates a stable ability to control the layout of multiple objects compared to TraDiffusion.

**Quantitative Results.** We compare our method with previous trajectory-based approaches on the proposed simple and complex trajectory tasks. As shown in Table 1, on the simple trajectory dataset, our method outperforms TraDiffusion in both DTL and IOT. However, since simple trajectories are typically single curves, the improvement in IOT is quite limited. The difference becomes more pronounced on the complex trajectory dataset. This demonstrates that our approach effectively aligns control trajectories and provides fine-grained control over the object.

## 4.3 ABLATION STUDY

**Ablation Study of the Hierarchical Guidance Mechanism (HGM).** We conduct an ablation study on the components of the HGM, including the Layout Guidance Module over the 8x8 resolution cross-attention maps and the Shape Guidance Module (SGM) over the 16x16 and 32x32 resolution cross-attention maps. As shown in Table 4, indicating that the addition of each component improved both the DTL and IOT metrics, further validating the effectiveness of our design. Besides, we observe that as the control ability improves, the FID score decreases. However, this slight quality trade-off is worthwhile for achieving more precise object control. Additionally, since our ablation dataset only contains 1,500 images, we believe this gap will diminish as the dataset size increases. We additionally provide qualitative results of adding the SGM over 32x32 resolution upsampling layers. When finer control over the object is required, the representation capability of object shape details in the 16×16 resolution cross-attention maps remains limited. This limitation makes it easy

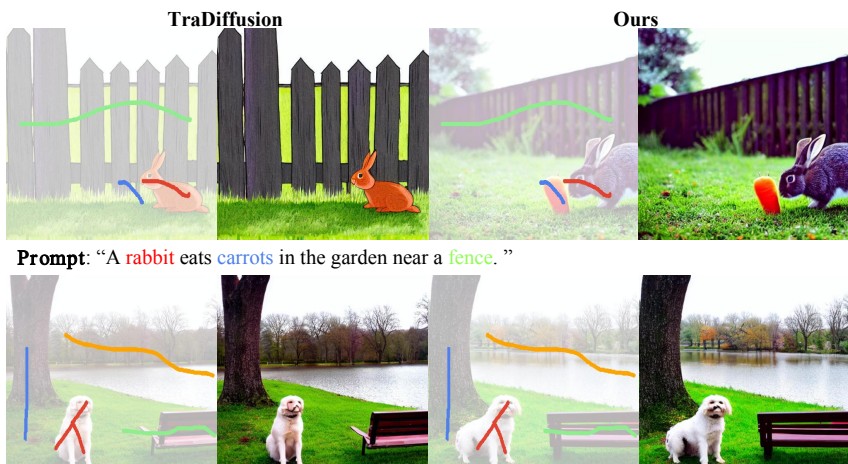

**Prompt**: "A rabbit eats carrots in the garden near a fence. "

**Prompt**: "A dog sits by a river under a tree, next to a bench."

Figure 10: **Visualization of multiple objects layout generation**

to overlook or misinterpret details during the model generation process, thus posing challenges for achieving fine control, as shown in Figure 7 (a). After adding 32x32 SGM, as shown in Figures 7 (b), the shape of the dog is better controlled.

**Ablation of the Fix Loss.** We conduct an ablation study on Fix Loss in the entire method. As shown in Table 3, the introduction of Fix Loss resulted in an improvement in DTL and IOT performance. This is because the initial attention map values can have different distributions under different text prompts and random seeds. Relying solely on the control loss makes it difficult to adequately cover the entire control area, which may result in the loss of control regions during the energy function guidance process, leading to issues of discontinuity in object generation and loss of details. As

Table 3: **Ablation of the Fix Loss.**

| methods → | w/o fix loss | ours |
|---|---|---|
| IOT(↑) | 0.63 | 0.67 |
| DTL(↑) | 0.0209 | 0.0214 |
| FID(↓) | 59 | 61 |

illustrated in Figure 9, during the guidance process, the middle part of the snake lacks attention, resulting in the generation of two similar snake-like objects. By introducing our Fix Loss, we can effectively focus on the parts that were overlooked during generation, ultimately producing a coherent snake that aligns with the trajectory.

## 5 LIMITATIONS

Although our method achieves fine-grained control of object generation based on trajectories, similar to previous work, it has only been tested on the SD-v1.5 version, and its transferability has not been further explored. Additionally, while our constructed complex trajectory dataset filters out some abnormal trajectories, further manual screening is still necessary. Moreover, we have observed that as control ability increases, the FID decreases. The underlying mechanisms behind this phenomenon require further exploration.

## 6 CONCLUSIONS

Based on the analysis of cross-attention maps in the Stable Diffusion generation process, we find that the model controls the generation of object layout at low resolutions, while at higher resolutions, it focuses on generating object shape. As the resolution increases, the details of the object's shape become clearer. Building on this finding, we improve previous work without the need for training and achieve fine control over the object through trajectories. Both qualitative and quantitative analyses demonstrate the effectiveness of our method. We hope that these insights into Stable Diffusion will inspire other tasks involving generation and editing.

## ETHICS STATEMENT

First, this research does not involve any experiments, surveys, or other interactions involving human subjects, thereby excluding ethical concerns related to such risks. We strictly adhere to the ethical guidelines established by the academic community as well as relevant laws and regulations, ensuring a high standard of ethics throughout the research process. Furthermore, the dataset constructed in this study will be made fully open after the research concludes to promote transparency, openness, and reproducibility in peer scientific research, aiming to contribute to the advancement of science. We also ensure that the dataset will not contain any information that could lead to privacy breaches or misuse, thereby maximizing data security and privacy. Throughout the research process, we strive to maintain fairness and impartiality, firmly opposing any form of bias or discrimination. Whether in the construction of the dataset or in the analysis of the research results, we have implemented rigorous measures to avoid potential biases and ensure equal treatment of all subjects. We adhere to the legal framework for research compliance, ensuring that every aspect of the study meets the requirements of existing laws and regulations, thereby maintaining the legitimacy and legality of the scientific inquiry. At the same time, we are committed to upholding research integrity to ensure the authenticity, objectivity, and scientific nature of the results, aiming to provide reliable theoretical and practical references for the related field.

## REPRODUCIBILITY STATEMENT

This study follows reproducibility principles, ensuring that the datasets, code, and experimental settings used are described in detail within the text. The source code and datasets for all experiments will be made available in publicly accessible repositories to allow other researchers to verify and reproduce our results.

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

## A    DATASET CONSTRUCTION DETAILS

**Filtered COCO2014 Dataset.** Following previous work Chen et al. (2024b); Wu et al. (2024), our dataset is constructed based on the COCO2014 training datasets Lin et al. (2014). First, we replace human-related vocabulary with the parent class "person" according to the caption annotations. Next, we filter based on whether the prompts contain plural nouns or multiple identical nouns. Then, using instance annotations, we filter out examples with bounding box areas smaller than 5% or larger than 80%, sorting them from largest to smallest. Finally, we select objects whose labels are included

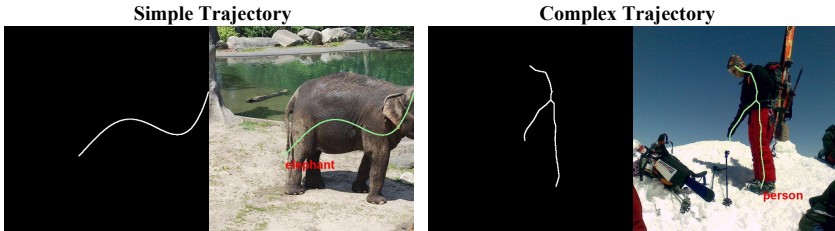

Figure 11: **Visualization of examples of simple and complex trajectories.**

Table 4: **Ablation of Different Losses in Attention Guidance over the 16×16 resolution cross-attention maps.**

| Guidance Component | 8x8 Layout Guidance Control Loss&Suppress Loss | 16x16 Attention Guidance Control Loss | Suppress Loss | Fix Loss | Metrics IOT($\uparrow$) | DTL($\uparrow$) | Fid ($\downarrow$) |
|---|---|---|---|---|---|---|---|
| 1 | ✓ | ✓ | ✓ | ✗ | 0.56 | 0.0230 | 66 |
| 2 | ✓ | ✓ | ✗ | ✗ | 0.62 | 0.0209 | 59 |
| 3. | ✓ | ✓ | ✗ | ✓ | 0.65 | 0.0212 | 60 |

in the prompts, storing the object masks with a maximum of 3 objects to create the foundational dataset.

**Simple Trajetory Datasets.** Following previous work, we generate a simple trajectory for the object by fitting a curve using polynomial regression based on the object masks. As shown in Figure 11, we randomly select 1,000 images to create the simple trajectory dataset.

**Complex Trajetory Datasets.** The simple trajectories are insufficient to effectively represent the shape of the objects. Therefore, we use the "morphology.skeletonize" function from the Python Skimage Van der Walt et al. (2014) library to extract the skeletons of the objects. However, the extracted results are too detailed, containing excessive branches. We iteratively remove the smallest branches, ultimately retaining 1 to 2 main branches to represent the complex trajectories of the objects, as shown in Figure 11. Similarly, we randomly select 1,000 images to create the complex trajectory dataset.

For the ablation experiments, we utilize a dataset consisting of 500 images sequentially selected from the simple trajectory datasets, combined with the 1,000 images from the complex trajectory datasets, resulting in a total of 1,500 images for qualitative evaluation.

# B ABLATION STUDY

**Ablation of Different Losses on 16×16 Resolution Attention Guidance.** We conduct an ablation study of different losses, including Control Loss (CL), Suppress Loss (SL), and Fix Loss (FL), in Attention Guidance over the 16x16 resolution cross-attention maps. As shown in Table 4 (1), under the control of the 16×16 resolution cross-attention map, although using CL and SL achieves the highest DTL, its IOT performance is the worst. This is because DTL can only measure the adherence of the generated object to the trajectory and cannot effectively evaluate whether the object correctly follows the trajectory when the generation is abnormal. As mentioned in Section 3.3.1, under the influence of SL, the generated object tends to overfit the trajectory; however, there are significant differences between the trajectory control region and the object's shape, resulting in distortion of the generated object. By removing the SL, IOT significantly improves, but DTL correspondingly decreases. Furthermore, as noted in Section 3.3.1, there is an issue of control region loss during the guidance process of the energy function. By incorporating our designed FL, both DTL and IOT are improved, which also indirectly verifies the effectiveness of our design.

**Ablation of Suppress Loss (SL).** We conduct an ablation study of SL in Layout Guidance over the 8x8 resolution cross-attention maps. As shown in Figure 12, SL effectively addresses the problem of chaotic object generation and improves stable layout control. At the same time, the generated objects do not excessively fit the provided trajectory. This is because the 8×8 cross-attention maps only

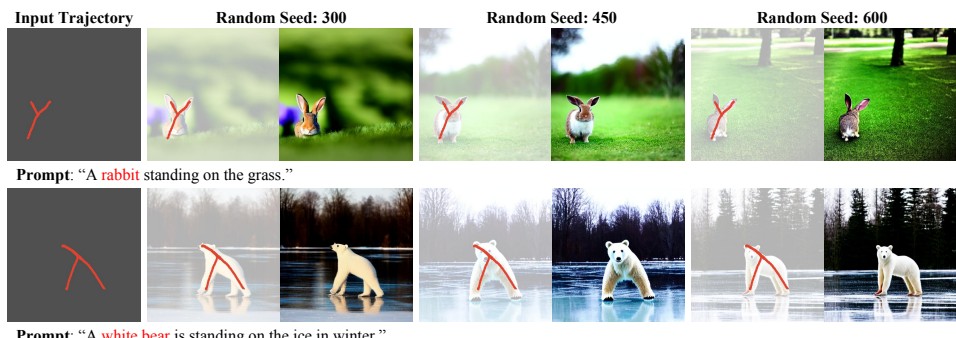

**Input Trajectory**     **Control Loss**     **Control Loss and Suppress Loss**

**Prompt**: "A dog is standing on the grass."

**Prompt**: "An elephant is walking down the street."

Figure 12: **Qualitative results of ablation study on Suppress Loss**

**Input Trajectory**     **Random Seed: 300**     **Random Seed: 450**     **Random Seed: 600**

**Prompt**: "A rabbit standing on the grass."

**Prompt**: "A white bear is standing on the ice in winter."

Figure 13: **Qualitative results of object fine-grained generation with different random seeds.**

correspond to the layout of the final generated object and cannot represent object shape details. As the resolution of cross-attention maps increases and the denoising process iterates, Stable Diffusion progressively refines the shape details of the objects. As shown in Table 12, with SL, IOT, DTL, and FID all show better performance.

## C  APPLICATIONS AND QUALITATIVE RESULTS.

**The Impact of Different Random Seeds.** As shown in Figure 13, our method can achieve stable fine-grained control of objects based on trajectories under different random seeds.

**The Impact of Prompt Complexity.** Since our method controls the cross-attention maps corresponding to the object tokens, we investigate whether our method can still achieve fine-grained control of objects as the complexity of the prompts increases and the cross-attention maps become more complex. As shown in Figure 14, under complex prompts, we can still achieve trajectory-based fine-grained control of objects while retaining other information from the prompts. In contrast, TraDiffusion does not possess this capability.

**Qualitative Results of Controllable Image Generation Experiments on the COCO2014 Dataset.** We additionally present the qualitative results of Table 1 on the COCO 2014 dataset, as shown in Figure 15. Our method achieves stable control of object layout generation and fine-grained control under complex trajectories.

**Multiple Objects Layout Generation.** As shown in Figure 16, our method can stably control the layout generation of multiple objects simultaneously, while TraDiffusion has shortcomings in this regard.

Table 5: **Ablation of Suppress Loss in Layout Guidance over the 8×8 resolution cross-attention maps.**

| Guidance Component | 8x8 Layout Guidance | | Metrics | | |
| | Control Loss | Suppress Loss | IOT(↑) | DTL(↑) | Fid (↓) |
| --- | --- | --- | --- | --- | --- |
| 1 | ✓ | ✗ | 0.31 | 0.0062 | 66 |
| 2 | ✓ | ✓ | 0.50 | 0.0098 | 63 |

**TraDiffusion**       **Ours**

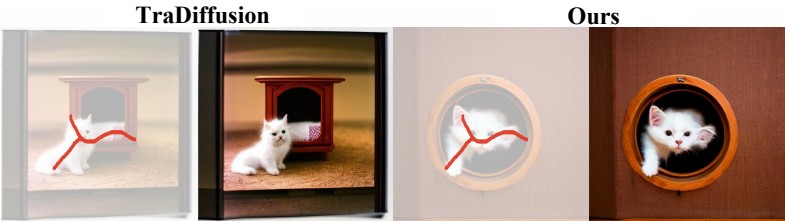

**Prompt:** "beautiful white kitten in a dog house, studio photography, high resolution, Cinestill 50, clear focus, Mamiya RZ67, 35mm photograph, Ultra-HD, wildlife photography, day light, high detail, complex details, Sony Alpha 7, ISO800, clear focus, soft lighting, super detailed, Sony Alpha 7, 8K --upbeta --v 4"

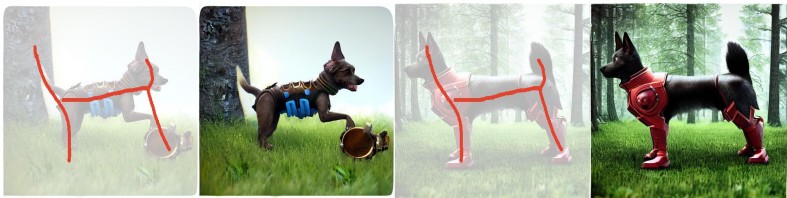

**Prompt**: "Super cute dog warrior wearing future armor photorealistic, 4K, ultra detailed, vray rendering, unreal engine --q 2"

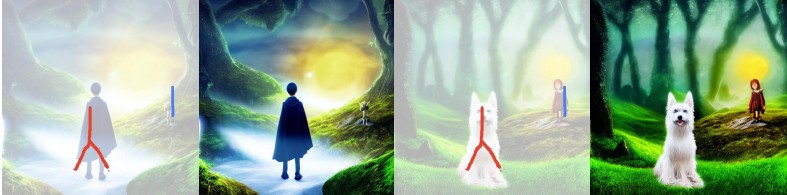

**Prompt**: "a young wizard stands at the edge of the abyss in a magical world, fairy forest, fairy mountains, super realistic style, fairy tale, white dog next to the wizard, white sun, early morning"

Figure 14: **Qualitative results of object fine-grained generation with complex prompts.**

**Multiple Objects Fine-Grained Generation.** We additionally present qualitative results of fine-grained control of multiple objects based on trajectories, as shown in Figure 17. TraDiffusion not only fails to achieve fine control of objects but also cannot stabilize the generation of object layouts. In contrast, our method demonstrates excellent control capability.

**Single Token Controlled by Multiple Trajectories.** As shown in Figure 18, our method can effectively distinguish multiple trajectories and generate multiple objects while achieving stable control of object layouts and fine-grained generation.

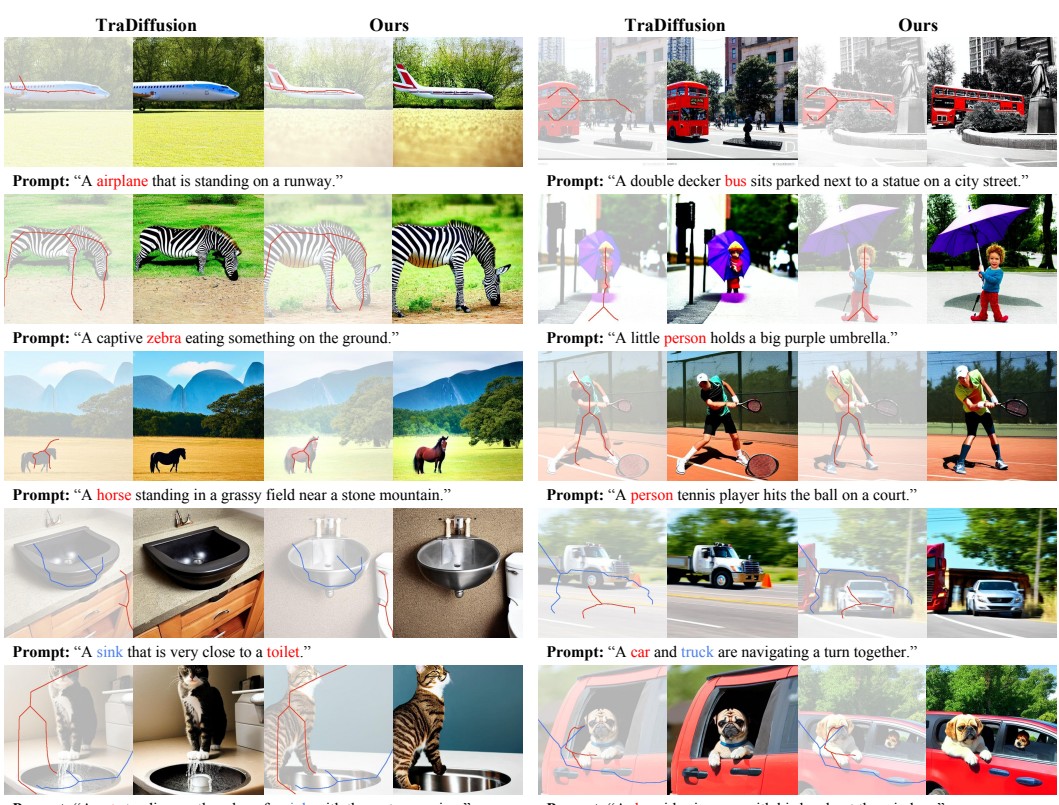

Figure 15: **Qualitative results of controllable image generation experiment on the COCO2014 dataset.**

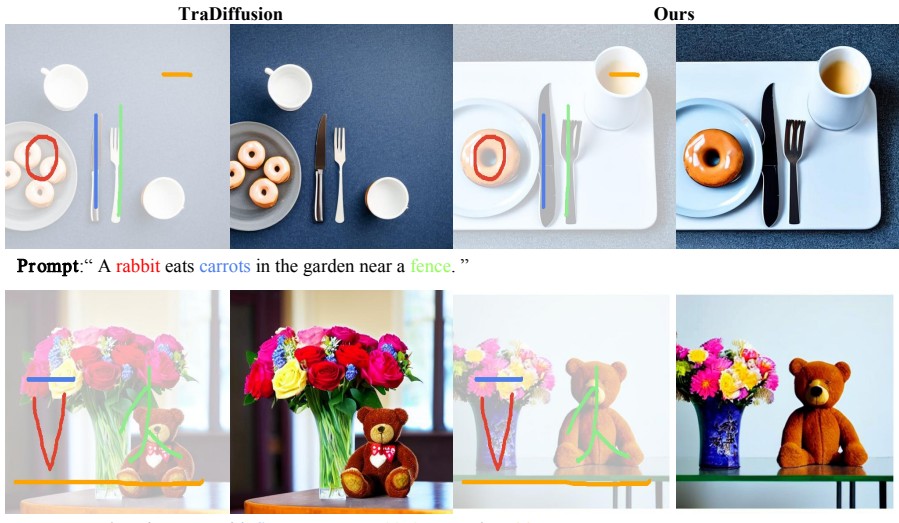

Figure 16: **Visualization of multiple objects layout generation**

**TraDiffusion**                    **Ours**

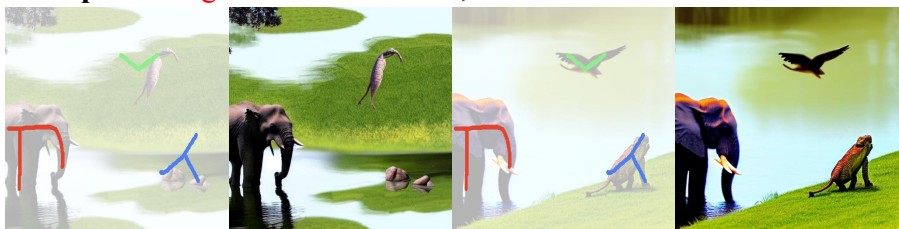

**Pompt**: "A dog stands under a tree, and a bird flies in the forest."

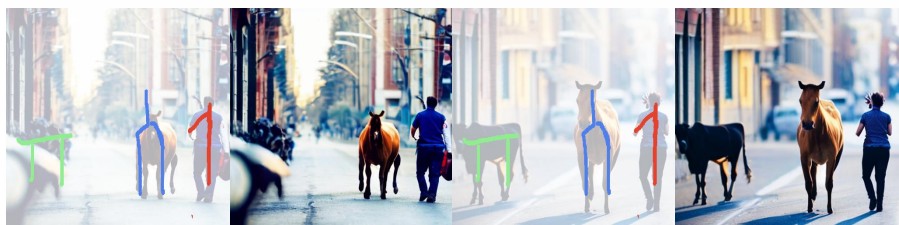

**Pompt**: "An elephant stands by a pond, with a lizard resting on the bank and a bird flying overhead."

**Pompt**: "A person is leading a horse near a cow walking down the street."

Figure 17: **Visualization of multiple objects fine-grained generation**

**TraDiffusion**              **Ours**

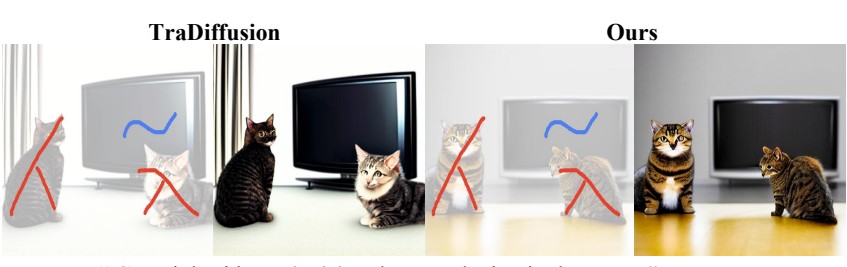

**Prompt**:" Cats sit beside a television that are playing in the room. "

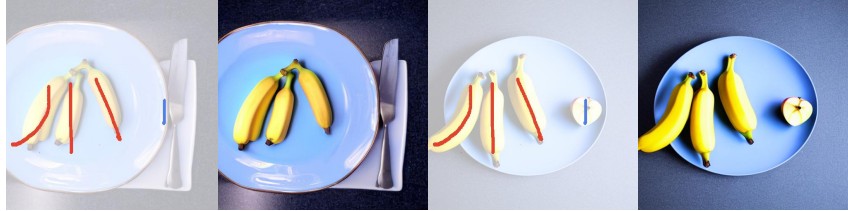

**Prompt**:" There are some bananas and an apple on the plate."

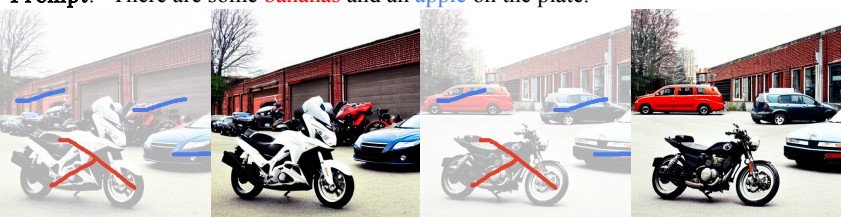

**Prompt**:" a motorcycle beside some cars. "

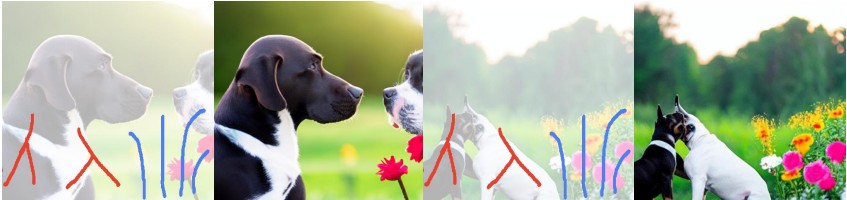

**Prompt**:" There are two dogs playing with each other near some flowers. "

Figure 18: **Visualization of Single token controlled by multiple trajectories.**

