# OpenReview forum: "Tradiffusion++：Hierarchical Guidance for Fine-Grained Trajectory-Based  Image Generation"
_ICLR.cc/2025/Conference — ICLR 2025 Conference Withdrawn Submission_

### Official Review · Reviewer_mFuw · 2024-10-31

**Soundness:** 2
**Presentation:** 2
**Contribution:** 1
**Rating:** 3
**Confidence:** 5

**Summary:**

The paper introduces an enhanced version of TraDiffusion that enables both coarse and fine control over image generation using input trajectories.
TraDiffusion offered only coarse control by focusing on low-resolution cross-attention maps, limiting its capability to follow complex trajectories.
To address this, the paper presents a Hierarchical Guidance Mechanism (HGM) with three guidance losses to facilitate fine-grained trajectory control.
By considering different cross-attention resolutions, the proposed approach can control both object layout and shape.
For evaluation, the paper proposes a dataset and metric specific to trajectory control.
Experiments demonstrate that the approach effectively manages complex trajectories and yields stable results across different seeds.

**Strengths:**

- The paper is easy to follow and the proposed method is clearly explained.

- Several figures were provided to illustrate different components of the pipeline in details.

**Weaknesses:**

- I generally find "trajectories" less intuitive and user-friendly than "masks," particularly for articulated objects like humans or animals. Drawing a trajectory or skeleton for such objects requires some level of artistic skill to envision a natural pose, which can be challenging.
On the other hand, masks don’t impose strict control on the model, giving it the freedom to generate a natural appearance for the object.
For example, how would one know that the "lambda" shape in Figure 6 is suitable for a cat? Similarly, the teddy bear in Figure 8.
It is evident in Figure 15 (the kid and bus) that unintuitive trajectories can lead to distorted objects. I believe that the usability of trajectories needs to be investigated compared to masks through a user-study.

- The main contribution, consisting of the three losses, is relatively minor and heavily inspired by [1]. The main difference is using a trajectory mask rather than bounding boxes (rectangular masks) to optimize different cross-attention maps. In fact, at low-resolution layers, it becomes identical to that of [1] for small trajectories, as illustrated in Figure 5. Therefore, I find the technical novelty of the proposed approach quite limited.

- The insights mentioned in section 3.2 are not new and were discussed in detail in several earlier works such as [2-4]. How do the insights provided in the paper differ from those in [2-4]?

- The "Coordinate Transformation" operation in Figure 3 dilates the trajectories, making them closely resemble segmentation masks. Consequently, a comparison with segmentation-based approaches, such as those in [5,6], is necessary to determine whether "trajectories" offer any advantage over segmentation maps.

[1] Chen, Minghao, Iro Laina, and Andrea Vedaldi. "Training-free layout control with cross-attention guidance." Proceedings of the IEEE/CVF Winter Conference on Applications of Computer Vision. 2024.

[2] Hertz, Amir, et al. "Prompt-to-prompt image editing with cross attention control." arXiv preprint arXiv:2208.01626 (2022).

[3] Tang, Raphael, et al. "What the daam: Interpreting stable diffusion using cross attention." arXiv preprint arXiv:2210.04885 (2022).

[4] Liu, Bingyan, et al. "Towards Understanding Cross and Self-Attention in Stable Diffusion for Text-Guided Image Editing." Proceedings of the IEEE/CVF Conference on Computer Vision and Pattern Recognition. 2024.

[5] Yunji Kim, Jiyoung Lee, Jin-Hwa Kim, Jung-Woo Ha, and Jun-Yan Zhu. Dense text-to-image
generation with attention modulation. In Proceedings of the IEEE/CVF International Conference
on Computer Vision, pp. 7701–7711, 2023.

[6] Guillaume Couairon, Marlene Careil, Matthieu Cord, Stephane Lathuiliere, and Jakob Verbeek. ´
Zero-shot spatial layout conditioning for text-to-image diffusion models. In Proceedings of the
IEEE/CVF International Conference on Computer Vision, pp. 2174–2183, 2023.

**Questions:**

- A comparison against segmentation-based approaches would be beneficial in finding use cases where trajectories can be useful over segmentation masks. This can simply be done by dilating the trajectories to form segmentation masks.

- Some generated images in Figure 8 and Figure 15 look unnatural/distorted, such as the surfer and the teddy bear. What is the reason for that?

- What is the reason for the degraded FID when different losses are added?

- In equation (4), is the denominator needed?

- IOT is not clearly explained. You mention that you predict a mask with Yolo. Do you compute the IOU between the trajectory and the segmentation mask? If yes, why do you think this is a suitable metric for evaluating abidance to the provided trajectories?

- Why is the guidance function computed 5 times during inference?

---

### Official Review · Reviewer_dKRb · 2024-11-03

**Soundness:** 3
**Presentation:** 3
**Contribution:** 2
**Rating:** 5
**Confidence:** 3

**Summary:**

This paper introduces training-free TRADIFFUSION++, aiming to solve the problem of TraDiffusion that cannot handle complex trajectories. A hierarchical guidance mechanism is designed, including layout guidance for low-resolution control, and shape guidance for high-resolution shape refinement. An IoT metric is introduced to evaluate the trajectory-based generation.

**Strengths:**

1. This paper carefully studied different attention layers in the UNet, proposing corresponding guidance, low resolution layout control, and high resolution shape control.
2. Control Loss and Suppress Loss is designed for layout guidance, and Fix Loss for shape guidance.
3. Extensive ablations are conducted to verify the effectiveness.

**Weaknesses:**

1. More elaboration of control loss compared to [1] is needed. What is the difference?
2. More qualitative ablation studies can be done, using the same text and trajectory prompt, to verify the effectiveness.
3. More visulization results are expected, such as more complex trajectories with overlapping

[1] Chen, Minghao, Iro Laina, and Andrea Vedaldi. "Training-free layout control with cross-attention guidance." Proceedings of the IEEE/CVF Winter Conference on Applications of Computer Vision. 2024.

**Questions:**

1. What is the difference between the control loss and the backward guidance in [1]?
2. When conducting qualitative ablation, can the authors use the same text and trajectory prompt to ensure the soundness?


[1] Chen, Minghao, Iro Laina, and Andrea Vedaldi. "Training-free layout control with cross-attention guidance." Proceedings of the IEEE/CVF Winter Conference on Applications of Computer Vision. 2024.

---

### Official Review · Reviewer_DBb7 · 2024-11-04

**Soundness:** 3
**Presentation:** 3
**Contribution:** 2
**Rating:** 5
**Confidence:** 5

**Summary:**

The authors introduce TraDiffusion++, which incorporates hierarchical guidance to enable fine-grained, trajectory-based image generation. This approach significantly improves upon the previous TraDiffusion model, particularly in terms of shape and layout accuracy. Compared to other kinds of methods like ControlNet, it is a training-free method so can reduce  the need for extensive training. Furthermore, TraDiffusion++ supports not only single-object generation but also multi-object generation.

**Strengths:**

The method is a simple and effective training-free method, which is quite good. The paper is also well-written and easy to understand. Experiment results also show that it indeed improves the previous method especially comparing to Tradiffusion.

**Weaknesses:**

I mainly have concerns of the experiment comparison and the discussion of a series of training-free methods.

Experiment comparison. The authors mainly treat TraDiffusion as a baseline, a training-free based method. However, it is still necessary to compare with other popular baselines like ControlNet and InstanceDiffusion etc. Despite they are not training-free methods, the ControlNet is already well-developed enough and very easy to use when just input the layout/skeleton images. The authors can also consider making them integrated into ControlNet to show the orthogonal ability of the proposed method. Moreover, the authors also should compare with other training-free methods especially leveraging the guidance such as [1] or some feature injection methods [2]. I recommend the authors do a more comprehensive comparison to other training-free methods.

More importantly, the authors point out a good point that it is an energy function method (Line 185). I suggest the authors could elaborate more about what the differences between energy-based methods and the other gradient-based method [1], and the advantage compared to feature injection methods like ControlNet and [2] etc. The authors can consider to analog the diffusion generation to the SGD, which is pointed out in [3]. These discussion will make the paper more theoretically-sound.

[1]. Universal Guidance for Diffusion Models
[2] Plug-and-Play Diffusion Features for Text-Driven Image-to-Image Translation
[3] Zero-to-Hero: Enhancing Zero-Shot Novel View Synthesis via Attention Map Filtering

**Questions:**

See the weakness parts. Besides, since it is a training-free method, I think it is very necessary for the authors to conduct more experiments on better diffusion models. I do not think the extra cost would be high.

---

### Official Review · Reviewer_2bAd · 2024-11-05

**Soundness:** 2
**Presentation:** 4
**Contribution:** 3
**Rating:** 6
**Confidence:** 4

**Summary:**

Tradiffusion++ proposes several techniques to amend issues in the Tradiffusion paper. The paper focuses on a single style of control mechanism (via scribbles) for training-free controlled generation of diffusion models. Compared to Tradiffusion, the paper proposes three types of guidances under different resolution and stages of the Diffusion models. Qualitative figures are provided to support the claims of effects of each proposed components.

**Strengths:**

- **Good presentation.** The paper is pretty well-written and easy to follow with qualitative figures and motivations introduced for each component in the paper. Each component seems reasonable contributions and innovations over the main baseline (i.e., TraDiffusion), which justifies the technical contributions of the paper.
- **Good potential applications.** Existing work mostly focus on controlling Diffusion models to generate images following semantic labels, depth, and canny edges. These representations tend to require more costly labor to acquire compared to scribbles. Hence, this line of research has potential practical value in real applications.
- **Good ablation experiments.** Extensive quantitative and qualitative evidence are used to justify the contributions of each individual module in the paper, which again shows the improvement over TraDiffusion for this paper to be a standalone work.

**Weaknesses:**

- **Insufficient experiments.** The main weakness of this version of the paper is the lack of comparisons with existing work. The main table (Table. 1) only compares the paper to TraDiffusion and plain stable diffusion, but no other types of controllable diffusion generation methods. There are immediate questions of how TraDiffusion++ compare to these baselines
    - Control methods that require training such as ControlNet and ControlNext. Can these methods be adapted for scribble-based generation?
    - Training-free methods such as [A] support generation with bounding boxes. Can they be adapted? Even if the answer is not, one very intuitive baseline is to use a bbox to bound the scribbles, and run the generation with the bounding boxes.
- **More diverse generation examples.** Though the proposed method seems generic and the paper includes many qualitative samples. They focus primarily on generation of scenes involving animals. Inclusion of other objects may be interesting.


[A] Chen, Minghao, Iro Laina, and Andrea Vedaldi. "Training-free layout control with cross-attention guidance." Proceedings of the IEEE/CVF Winter Conference on Applications of Computer Vision. 2024.

**Questions:**

The detailed questions are listed in the weaknesses section, but here is the summary:

- Can the author provide results (even a small-scale one is ok) for comparisons with existing baselines for more results on Tab. 1?
- Inclusion of more diverse qualitative examples.

---

### Note · Authors · 2024-11-14

I have read and agree with the venue's withdrawal policy on behalf of myself and my co-authors.